# The Effect of Molten Phosphate on Corrosion of 316 Stainless Steel, Alloy 625, and Titanium TA8 in Supercritical Water Oxidation Conditions

**DOI:** 10.3390/ma16010395

**Published:** 2023-01-01

**Authors:** Zitao Lin, Pengwei Ren, Riguang Zhu, Xingying Tang, Youchang Zheng, Tiantian Xu, Yinghui Wang, Jianqiao Yang, Jianjun Cai

**Affiliations:** 1Guangxi Key Laboratory on the Study of Coral Reefs in the South China Sea, School of Marine Sciences, Guangxi University, Nanning 530004, China; 2School of Resources, Environment and Materials, Guangxi University, Nanning 530004, China; 3School of Energy and Power Engineering, Xi’an Jiaotong University, 28 Xianning West Road, Xi’an 710049, China; 4School of Architecture and Traffic, Guilin University of Electronic Technology, Guilin 541004, China

**Keywords:** supercritical water oxidation, stainless steel, nickel-based alloy, titanium alloy, corrosion, phosphates

## Abstract

The early formation of phosphate oxide formed on 316 stainless steel (316 SS), nickel-based Alloy 625, and titanium alloy TA8 exposed in supercritical water (400 °C, 25 MPa) containing phosphate, chloride, and oxygen was investigated. Phosphate corrosion products of austenitic stainless steel displayed the severest spallation. Stable phosphates oxide films were inclined to form on Alloy 625. TiO_2_ and Ti_2_O_3_ are the two main components of oxide films on TA8. There is a strong synergistic effect between phosphates, oxygen, and supercritical water, leading to severe corrosion. The corrosion behavior of the three alloys at the top and bottom of the reaction tube was compared. Both at the top of the reaction tube and at the bottom of the reaction tube, TA8 showed an increase in mass. 316 SS and alloy 625 showed mass gain at the top and mass loss at the bottom. The alloys’ detailed molten corrosion mechanism after exposure to supercritical water is discussed.

## 1. Introduction

Supercritical water (SCW) is defined as the water above the critical point (T = 374.3 °C, P = 22.1 MPa), which shows some unique physical properties such as weak hydrogen bonding, high diffusion coefficient, and low viscosity. Under appropriate conditions, supercritical water oxidation (SCWO) can have conversion efficiencies higher than 99.9%. At the same time, the organic matter is entirely wholly degraded into harmless small molecules (i.e., H_2_O, CO_2_, N_2_, and inorganic salts) in a short reaction time. Therefore, SCWO is generally considered an eco-friendly technology for treating high-concentration organic wastewater and sewage sludge, indicating that this technology has broad development prospects and commercialized value [1]. Moreover, supercritical water oxidation technology can transform organic matter into hydrogen, oil, and gas [2]. However, raw materials contain corrosive reaction components such as aggressive ions (e.g., SO_4_^2−^, Cl^−^, F^−^, and PO_4_^3−^), and oxygen would cause severe corrosion under high pressure and high-temperature conditions. Hence, the corrosion problem is the main obstacle to commercializing supercritical water and supercritical water oxidation technologies. [3,4,5,6]. Many scholars are devoted to the research and development of various methods to solve corrosion problems in supercritical water oxidation technology, such as developing new reactors, maintaining the composition and concentration of the reactant feed, and researching new anti-corrosion materials [7,8,9,10]. However, no single material or method has been found to cope with all corrosion problems in supercritical water oxidation environments. To assess the extent to which equipment materials are subjected to corrosion in actual industrial production and to ensure the smooth operation of SCWO systems, many materials have been selected for corrosion testing in the specific corrosive environments associated with SCWO. Because of their advantages in mechanical strength, corrosion resistance, and manufacturing cost, austenitic stainless steel, nickel-based alloy, and titanium alloy are often three typical candidate alloys for the construction of SCWO system equipment [11,12,13,14,15].

In corrosion tests under subcritical and supercritical conditions, 316 SS always showed weak corrosion resistance and high corrosion rates, and the corrosion increased with increasing temperature. A double layer of oxide was formed on the surface of 316 SS, with an outer layer of iron-rich oxide and an inner layer of Fe/Cr/Ni-rich oxide. Nickel-based alloys are used for their excellent corrosion resistance and high strength in harsh conditions in chemical, coal-fired, and nuclear power plants. In SCWO systems, nickel-based alloys are also widely used to manufacture reactors and heat exchange tubes. In supercritical water oxidation corrosion tests, the oxide film on the surface of nickel-based alloys usually behaves as a double oxide layer, consisting of a loose outer layer of NiO, Fe oxides, and inorganic salt deposits and an inner layer consisting of a compact inner layer of internal NiO layers rich in chromium oxides. However, nickel-based alloys are poorly resistant to corrosion in supercritical aqueous NaOH solutions because the protective metal hydroxides on the material’s surface can be melted at supercritical temperatures. In SCWO application, titanium alloys have better corrosion resistance than nickel-based and iron-based alloys and are usually not corroded by HCl solutions at high temperatures but show very little corrosion resistance to H_3_PO_4_ solutions above 400 °C [16]. It should be noted that 316 SS, Alloy 625, and TA8 are often used as candidate alloys for different component areas of supercritical water oxidation systems. Three other alloys have been chosen to assess the corrosion behavior of components in various areas of supercritical water oxidation systems in a hydrothermal phosphate molten salt solvent system.

The phosphate was proven to reduce corrosion due to the formation of passivation metal phosphating film [15]. Ma et al. [17] proposed that phosphate film inhibited further corrosion of the alloy by isolating corrosive substances from the alloy substrate. Kritzer et al. [18] found that a secondary passivating metal phosphate film formed on the surface of alloys, improving corrosion protection. Ou et al. [19] studied that stable phosphate oxides were observed on the surface of nickel-based alloys in supercritical water. However, corrosive substances also destroy the water-insoluble phosphate film [14]. Phosphate would be hydrolyzed into hydrogen phosphate, which owns a low melting point, leading to the formation of mixed molten salt in the supercritical water [20,21,22]. It was found that metal oxides could dissolve into the ionic melts of low-melting phosphates to form the eutectic FePO_4_—Ni_3_(PO_4_)_2_—Na_3_PO_4_—Na_2_HPO_4_ molten salt, composing the outer layer of oxide film [23]. In a recent study, Voisin et al. [24] reported the discovery of hydrothermal molten salt (HyMoS). They investigated the use of low-melting salts in SCWO systems to dissolve and remove the formed salt deposits, which can solve the problem of salt deposits in supercritical water oxidation technology. Although many previous studies have investigated the influence of aggressive ions such as chloride and sulfate, insufficient data is available on phosphate corrosion in supercritical water. However, few studies have reported the corrosion of material alloys in supercritical water oxidation systems containing molten phosphate salts. The phosphorus-containing wastewater and wastes widely come from pesticides, fertilizers, and industrial sludge. When treating organophosphorus waste by supercritical water oxidation, the reaction solution containing phosphorus and chlorine can cause corrosive effects on the alloy. Hence, molten phosphate corrosion should be paid enough attention in supercritical water. On the one hand, molten salt would isolate the substrate from aggressive species; on the other hand, oxide film would be damaged after dissolving into low-melting phosphates. In supercritical water, the effect of phosphate on alloy corrosion is complicated, so systematic research should be conducted to explore at present.

In this study, experimental research investigated the corrosion characteristics and mechanisms of stainless steel, nickel-based alloys, and titanium alloys in supercritical water conditions containing phosphate, chloride, and oxygen at 400 °C/25.0 MPa. This work’s objectives were to reveal phosphate’s effect on the corrosion of three typical alloys in supercritical water. The database obtained could contribute to the theoretical basics of corrosion prevention of SCWO.

## 2. Materials and Methods

### 2.1. Material

316 SS, Alloy 625, and TA 8 were selected as the tested alloys, and their main chemical compositions of specimens are listed in Table 1. The specimens were processed into 6 mm × 5 mm × 2 mm, and each coupon had a small hole securing it in the reaction tube. The specimens were mechanically polished to 2000 finish, degreased ultrasonically in acetone, and dried by a vacuum drier.

All the chemicals used in this experiment were purchased from Tianhao Chemical Co., Ltd. (Tai’an, China). NaCl (≥99.5 wt.% purity) and Na_3_PO_4_ (≥99.5 wt.% purity) were used as the corrosive species. The 30 wt % H_2_O_2_ of the solution was used as an oxidant environment.

### 2.2. Apparatus and Procedures

The corrosion test conditions were 400 °C/25 MPa with different exposure times. The corrosive solution contains Na_3_PO_4_, NaCl, and dissolved oxygen. Corrosion tests are performed in a batch reactor. Figure 1 shows the schematic diagram of the supercritical water experimental apparatus. The reactor was made of 304 stainless, and the maxing condition was 500 °C, 30 MPa. The alloy sheet is placed inside the reactor at room temperature. The reactor is purged with nitrogen and sealed after injecting the corrosion solution and hydrogen peroxide into the reactor. The reactor was vertically fixed and heated in a Fluidized Sand Bath. The reactor had a volume of 20 mL, and type-K thermocouples were applied to measure the reactor wall temperature. Temperature signals from the inner furnace and reactor wall were sent to a PID controller to regulate the target. The temperature-controlled precision is ±1 °C by a temperature controller. And then calculate the injected corrosion solution and hydrogen peroxide content based on temperature, pressure, and reactor volume parameters. The corrosion test parameters are shown in Table 2. The corrosion behavior of the tested alloys at the top and bottom of the reactor was investigated. The bottom specimens were immersed in a corrosive medium, and the top samples were secured in the air. After the corrosion test, the obtained specimens were cleaned with deionized water, ultrasonically degreased, cleaned with deionized water and acetone, respectively, and then dried by a vacuum drier. The specimens were weighed before and after exposure using an Electronic analytical balance (AUW220D) with a sensitivity of 0.1 mg.

### 2.3. Analysis

The morphologies of the tested alloys were observed by metallographic microscope (Nikon, LV150N, Tokyo, Japan). The surface and cross-section morphologies of the specimens were observed by a scanning electron microscope (SEM, HITACHI, Tokyo, Japan) equipped with the model SU5000. The content of elements in the corroded area of the specimens was analyzed using the energy dispersive spectrometer (EDS, Bruker, XFlash Detector, Billerica, MA, MA, USA). The X-ray diffraction (XRD, RIGAKU, Ultima IV, Akishima, Japan) instrument identified the oxide crystal structures. The oxide film’s chemical compositions were detected using an X-ray photoelectron spectrometer (XPS, Thermo Fisher, ESCALAB 250Xi, Waltham, MA, USA).

## 3. Results

### 3.1. Surface Morphologies

Figure 2 shows the images of the tested alloys exposed to SCW (400 °C/25 MPa) at experiment Run-6 for 200 h. At the top of the reaction tube, the oxide scales formed on the surface of specimens presented dense and continuous, indicating general corrosion. However, the bottom specimens showed different corrosion morphologies from the top specimens. The bottom 316 SS displayed significant spallation of oxide scales and pitting corrosion, indicating the corrosion product was unstable. The unstable oxide scale would offer the diffusion channel for a corrosive medium, leading to accelerated corrosion. The discontinuous and stable corrosion products were generated on the surface of Alloy 625 and TA8.

The surface SEM morphologies of the specimens exposed to SCW (400 °C/25 MPa) at experiment Run-6 for 200 h, as displayed in Figure 3. The EDS analysis revealed the element distribution of corrosion products on specimen 2 (see Figure 3). Significant changes in the corrosion morphologies were observed in the conditions with or without salt. 316 SS unsalted specimens, the surface is dominated by continuous dense needle-like oxide, while the salted specimens’ surface is loose and easy to peel off the lamellar product. The unsalted specimens of Alloy 625 formed a continuous fine and dense oxide film, whereas the surface products of the salted specimens were discontinuous needle-like and lamellar oxides. TA8 unsalted specimens show a more continuous and finer surface oxidation product than salted specimens. 316 SS displayed the severest corrosion compared with Alloy 625 and TA8. Significant corrosion product spallation of 316 SS was observed, and the loosely distributed block oxide remained on the surface after ultrasonic cleaning. According to the EDS analysis, the nodular oxide located at the position of A2 is composed of Fe and O. The priority formation of the outer iron oxide film was explained by a higher diffusion rate of Fe [25]. Plate-like oxide scales were scattered on the surface of 316 SS. The compact and stable oxide film was hard to form on the surface of 316 SS, resulting in the matrix being exposed to a corrosive medium, and further corrosion would occur under unprotected conditions. As a result of the oxide film being dense and uniform, the Nickel-based alloy 625 exhibited excellent corrosion resistance in SCW without salt. The nickel-based alloy is generally a candidate material for a supercritical water oxidation system [14,26,27]. Dense and uniform corrosion products help improve the corrosion resistance of Alloy 625. The acicular oxide was observed on the bottom Alloy 625 after exposure to supercritical oxidizing water with salt. Based on the EDS results, the acicular oxide was mainly composed of Ni and Fe, indicating nickel oxide and iron oxide could be stable in supercritical water with molten salt [28]. Zhang et al. [29] also found that the acicular oxides formed on alloy 690TT were rich in Ni and Fe but poor in Cr. Yang et al. [30] thought that the transverse diffusion of the metal cation was limited, and the oxide could only be produced in the direction perpendicular to the surface of the alloy coupons. Compared with 316 SS, the oxide films formed on Alloy 625 and TA8 were much smaller particles and continuous and dense morphologies. In supercritical oxidizing water, block corrosion products with a regular geometric shape formed on the surface of TA8, indicating the corrosion resistance of titanium alloy was better than stainless steel [31]. The EDS analysis results of corrosion products formed on the surface of tested alloys are shown in Figure 3. On the surface of bottom TA8, block products (Figure 3C2) were mainly composed of Ti and O, indicating the titanium oxides were stable and the stable phosphate corrosion products were hard to generate [32].

### 3.2. Surface Chemical Analysis of Tested Alloys

Figure 4 illustrates the relative atomic content of the main elements compositions of the oxides film formed on the tested alloy exposed to a supercritical water corrosion environment at different times. Chlorine was not detected in the surface layer of the tested alloys, indicating that chlorine was difficult to form stable oxides or chloride in SCW [33,34]. The Fe content on the surface of 316 SS displayed no significant change compared to the matrix, illustrating that the stable iron oxides could generate to isolate the alloy from the corrosive medium. Moreover, the Fe content of the specimens at the top of the reaction tube is higher than that at the bottom because there are iron depositions at the bottom. The nickel content decreased significantly on the surface of Nickel-based alloys, and the nickel content of 316 SS was hardly detected in experiments Run-1 to Run-6. Ni is usually selectively dissolved on the alloy surface in SCWO containing chloride, indicating that the dissolution of unstable chloride would cause severe pitting corrosion [35]. In supercritical oxidizing water, the Cr content of 316 SS surfaces decreased as the oxygen content increased. A synergistic corrosion effect with chlorine and oxygen would cause the strong spalling tendency of oxide films.

The content of nickel in the surface oxide film showed a more significant reduction than the other elements (see Figure 4b). The selective dissolution of nickel occurs under supercritical water conditions has generally been proved [35,36]. After a different exposure time of the corrosion test, the Fe content of the Alloy 625 surface layer displayed a noticeable increase. Fe had a strong tendency to convert into stable Fe_3_O_4_, which would help prevent further corrosion. In addition, the stability of oxides is in the order of FeCr_2_O_4_ > Fe_3_O_4_ > NiO, and the solubility of elements is in the order of Mo > Ni > Fe > Cr [37,38]. Iron and chromic oxides were both considered stable corrosion products in supercritical water; however, nickel oxides tended to selective dissolution [34]. As shown in Figure 4b, a higher concentration of phosphorus was detected on the surface of Alloy 625 than 316 SS and TA8, indicating the stable phosphate was likely to generate on nickel-based alloy [15]. Due to salt deposition in SCW, the phosphorus was hardly detected on the top specimen. In supercritical water, oxygen plays a vital role in forming a surface oxide layer, and the unstable outer film accelerates the process [5]. Owing to the protective phosphate oxide films on the surface, corrosion of Alloy 625 was slight at SCW condition [18]. Moreover, the Fe content presented a remarkable increase in the outer film of nickel-based alloy, indicating that the iron oxides also tended to stabilize on the alloy surface in supercritical water.

According to the EDS and XRD analysis of the titanium alloy TA8, the P content and phosphate oxides were detected a on the surface. The research result of titanium alloy is different from previous studies. Lu et al. [39] found that the Ti_5_O_4_(PO_4_)_4_ was the main phase of titanium oxy-phosphate and titanium oxide while in chlorpyrifos SCWO medium at 450 °C and 25 MPa when the concentration of oxygen was about 15,000 mg/L. Lu et al. also found that the titanium oxide phosphates of π-Ti_2_O(PO_4_)_2_•2H_2_O and (TiO)_2_P_2_O_7_ were formed on TA3 after exposure in the phosphoric acid with 1.0 mol/L at 250 °C [32]. The (TiO)_2_P_2_O_7_ was also observed on TA10 in supercritical oxidizing water containing phosphate and sodium chloride when oxygen concentration was 78,200 mg/L. Contrasted with previous studies and this study, the high oxygen concentration contributed to generating titanium oxide phosphates. Generally, the corrosion resistance of titanium alloy is dependent on the titanium oxide passivation layer which displays excellent resistance to hydrothermal chlorine corrosion. The TiO_2_ oxide layer plays a major role in the passivity process. However, the area of deposition phosphate will be corroded locally due to the molten salt corrosion and lack of oxidization passivation. Therefore, pitting corrosion was also found on the surface of titanium alloy in high temperature water containing phosphate. The formation of titanium oxide phosphates had revealed that the failure of passivation layer and reduction of corrosion resistance. Besides, the chlorine could accelerate corrosion and promote the decomposition of oxide layer [40,41]. Noteworthy, a few Fe was detected on the surface of TA8 indicated Fe dissolved from reactor inner wall and migrated. The molten salt phosphate would offer a fast-transferring channel for dissolved ions in the environment. Yang et al. also found the dissolution-precipitation process was the predominant corrosion mechanism of alloys in supercritical oxidizing water containing chlorine salt at 400 °C [42].

### 3.3. Corrosion Products Analysis of Tested Alloys

To investigate the compositions of corrosion products, Figure 5 depicts the XRD analysis results of corrosion products formed on 316 SS, Alloy 625, and TA8 after exposure to experiments Run-1 to Run-6 for 25 h, 100 h, 150 h, and 200 h, respectively. The crystal structure of the oxides on the surface of 316 SS was composed of Fe_3_O_4_, Fe_2_O_3_, and Cr_2_O_3_. Guo et al. [43] also found that Fe_3_O_4_ and Cr_2_O_3_ were generated on the surface of 316L stainless steel exposed to 550 °C~600 °C supercritical water. A minor amount of oxide of NiO signal was detected on the surface of 316 SS, indicating that nickel oxides were unstable in supercritical water containing oxygen and chlorine. In this study, the phosphates (i.e., FePO_4_, CrPO_4_, and Ni_3_(PO_4_)_2_) was not detected on the surface of 316 SS. The reason can be explained that the presence of chlorine, the peak area of the alloy oxides, shows a significant reduction, indicating chloride could lead to the decomposition of the alloy oxide films. Three firm matrix peaks were observed on the surface of 316 SS in the experiment Run-1 for 25 h; however, the peaks of other oxides were weaker and slight corrosion took place in a short exposure time. With the exposure time increased to 200 h, the peak strength of some oxides (i.e., Fe_2_O_3_, Fe_3_O_4_, and Cr_2_O_3_) was stronger than the experiment Run-1 for 25 h. Overall, the intensity of XRD detection of peaks of Fe_2_O_3_, Fe_3_O_4,_ and Cr_2_O_3_ in corrosion products increased as reaction time.

From Figure 5b, the strong peaks of the matrix were found in Alloy 625, indicating that thin oxide scales were produced on its surface [25]. On the surface of Alloy 625, the corrosion products were mainly composed of Cr_2_O_3_ and NiCr_2_O_4_. The inner layer is formed of fine particles, continuously and densely distributed on the whole surface of the alloy. The excellent stability of these oxides is conducive to the oxidation resistance of Alloy 625 alloy at high temperatures. Little CrPO_4_, Ni_3_(PO_4_)_2,_ and Na_3_PO_4_ were detected on the surface of Alloy 625, which reached a good agreement with the EDS analysis results, suggesting water-insoluble phosphates were stably distributed on the alloy. The water-insoluble phosphates formed on the surface could effectively inhibit the corrosive media from entering the matrix, improving the corrosion resistance of Alloy 625 in SCWO. Except for three strong matrix peaks, the XRD spectra of Alloy 625 were smooth, and the oxide peaks were weak in the experiment Run-1 to Run-6 for 25 h, 100 h, and 150 h. Alloy 625 displayed excellent corrosion resistance even in supercritical water with high concentrations of salts and oxygen. As the exposure time reached 200 h, a small quantity of NiCr_2_O_4_, Cr_2_O_3_, and Fe_2_O_3_ was observed on Alloy 625. The formation heat of spinel increases with the concentration of chromium until the ideal NiCr_2_O_4_ or FeCr_2_O_4_ composition is reached, and the diffusion rate decreases due to the well-ordered structure of spinel [44]. In addition, the peaks of phosphates (i.e., CrPO_4_, Ni_3_(PO_4_)_2_, and Na_3_PO_4_) were more distinct than those of exposure time was 25 h, 100 h, and 150 h. After ultrasonic cleaning, the phosphates adhere to the surface of Alloy 625, and the stable phosphates film represents the passivation film. The phosphates detected on the oxide scales of Alloy 625 are almost all in the form of orthophosphate.

As shown in Figure 5c, the corrosion product components of TA8 are mainly made up of TiO_2_ and Ti_2_O_3_. The peak intensities and peak locations were changed as the exposure time increased. Titanium oxide exhibited more stability than other oxides in SCWO.

### 3.4. XPS Analysis

The chemical valence of elements on the oxide films formed on the tested alloys at the bottom of the reaction tube exposed to the experiment Run-6 for 200 h was further analyzed by XPS. The core level spectra of Ni 2p, Cr 2p, Fe 2p, Na 1s, Ti 2p, and P 2p, as shown in Figure 6. In accordance with the NIST (National Institute of Standards and Technology), the binding energy of Ni 2p_3/2_ of 316 SS was located at 854.4 eV and 861.2 eV, corresponding to NiO and Ni(OH)_2_, respectively [45]. Ni(OH)_2_ and NiO are oxides formed by metal cation reactions [19]. The binding energy of Ni 2p_1/2_ of 316 SS was located at 871.8 eV, signified to NiO. The XPS spectra peaks of Ni 2p of oxide scale formed on 316 SS were weak, and few Ni and nickel oxides were detected in EDS results and XRD, which was highly consistent with the EDS analysis and the XRD patterns. In this part of Cr 2p of 316 SS, the binding energy of Cr 2p_3/2_ was located at 576.1 eV and Cr 2p_1/2_ was located at 585.9 eV, representing Cr_2_O_3_ and NiCr_2_O_4_, respectively [45,46]. The Fe 2p spectrum revealed the division into Fe 2p_3/2_ and Fe 2p_1/2_ centered at 710.4 and 723.5 eV, indicating the existence of Fe_2_O_3_ and Fe_3_O_4_ in the oxide film, respectively.

Apparently, the XPS spectra peaks of Ni 2p of oxide films formed on Alloy 625 were stronger than 316 SS, which has in good agreement with EDS results. The binding energy of Ni 2p_3/2_ of Alloy 625 was located at 855.6 eV and 861.5 eV, signifying NiO. The binding energy of Ni 2p_1/2_ of Alloy 625 was located at 873.7 eV and 879.3 eV, corresponding to Ni(OH)_2_ and NiCr_2_O_4_, respectively. In this part of Cr 2p of Alloy 625, the peaks located at 576.1 eV and 585.9 eV belong to Cr 2p_3/2_, corresponding to Cr_2_O_3_ [46]. The binding energy of Cr 2p_1/2_ was located at 586.4 eV, attributed to NiCr_2_O_4_. There are two peaks of Fe 2p in Alloy 625, one is Fe 2p_3/2_ with a peak at 711.2 eV, and the other is Fe 2p_1/2_ of the binding energy at 723.5 eV, corresponding to Fe_2_O_3_ and Fe_3_O_4_, respectively_._ From the above, the binding energy located at 1070.9 eV was the characteristic peak of sodium phosphate. The P 2p_3/2_ with an obviously strong peak of the binding energy at 133.2 eV, which stood for CrPO_4_ and was observed by XRD spectra. In the Ti 2p core level spectra, the signal located at a binding energy of 457.8 eV was assigned to Ti 2p_3/2_, and 464.4 eV was assigned to Ti 2p_1/2_ in TiO_2_. The core level spectra of Na 1 s and P 2p were detected on the corrosion products formed on the surface of TA8, indicating titanium could react with phosphate anion [15]. According to the EDS and XRD results, Fe_2_O_3_ and Fe_3_O_4_ are the corrosion products formed on the surface of 316 SS. The corrosion products generated on Alloy 625 included Cr_2_O_3_, NiCr_2_O_4_, and phosphates (i.e., FePO_4_, CrPO_4_, and Ni_3_(PO_4_)_2_), while the TiO_2_ and Ti_2_O_3_ are corrosion products covered on TA8.

### 3.5. Gravimetry

Figure 7 exhibits the corrosion weight gains of 316 SS, Alloy 625, and TA8 after exposure to different conditions of SCWO. Two parallel specimens were used for the weight loss measurement of each alloy.

The fluctuation in weight is due to the competition between oxidation and pitting or peeling of the oxide films [47,48]. As shown in Figure 7a, 316 SS showed the highest corrosion weight gain of 3.7 mg/cm^2^ in the experiment Run-6 for 200 h. Figure 7b showed the highest corrosion weight gain of −3.1 mg/cm^2^. The 316 SS contains much less amount of nickel and molybdenum, leading to weaker corrosion resistance than Alloy 625. These results agree with the analysis of other researchers. Alloy 625 displayed more corrosion resistance than stainless 316 in supercritical oxidizing water containing salt [38]. The high corrosion weight gain of 316 SS was attributed to the oxide scales covered on the surface displaying unstable and severe spallation in harsh conditions, which is in accord with the corrosion morphologies (Figure 3). Due to the outstanding corrosion resistance, Alloy 625 and TA8 are more appropriate to be considered candidate materials for SCWO [49]. Chang et al. [28] found that the weight change of alloy 625 was slight in supercritical water with 8.3 ppm (by weight) oxygen at 400 °C and 500 °C. Hatakeda et al. [50] derived the corrosion rate of Hastelloy-C276 of 5 to 34 mmpy when treating 3-chlorobiphenyl in the SCWO system at 477 °C and 30 MPa by measuring the wall thickness. Without oxide removal, in Figure 7, all three alloys at the top and TA8 at the bottom exhibit an increase in mass, indicating a positive corrosion rate. The 316 SS and Alloy 625 located at the bottom exhibited a decrease in mass, indicating a negative corrosion rate. Through the corrosion time of 200 h of, the mass change rate of the specimen is derived, located at the top of the 316 SS corrosion rate of 0.203 mmpy. Alloy 625 is 0.182 mmpy, TA8 is 0.269 mmpy, located at the bottom of the 316 SS corrosion rate of 0.170 mmpy, Alloy 625 is 0.010 mmpy, TA8 is 0.115 mmpy. The corrosion rates of 316 SS and Alloy 625 located at the bottom were lower than those at the top because the overall mass of the bottom specimen showed a weight loss in terms of mass change, while the overall mass of the top specimen showed an increase due to the flaking of the oxide film on the surface of the specimen due to hydrothermal molten phosphate corrosion. Alloy corrosion weight loss can cause more corrosion failure of equipment materials compared to corrosion weight gain. It should be noted that TA8 exhibits an increase in mass for both the top and bottom specimens, and the rate of mass change is positive, which can be attributed to the thicker oxide film of TA8, the better stability of the oxide film, and the fact that it is less prone to peeling. In the test conditions in this paper, TA8 showed the most corrosion resistance, while 316 SS had the highest corrosion rate, with alloy 625 in between them.

### 3.6. Structure and Composition of Corrosion Layer

In order to investigate the transformation of the tested alloys elements on the surface exposed to various corrosion conditions, the surface morphologies and EDS line scan analysis of the primary elemental distribution in oxide film formed on 316 SS, Alloy 625, and TA8 after exposure in SCW at experiment Run-6 for 200 h was performed, as shown in Figure 8. In the profile analysis, the surface to be analyzed is the lower surface of the specimen at the bottom of the reaction tube. Except for the significant O content increase, Ni, Cr, and Fe content was reduced. The decrease in the content of nickel was the most obvious. The reason can be explained by the synergistic effect of chloride and oxygen on the nickel dissolution [15]. The decrease in Cr content can be attributed to the oxidant inducing a higher electrochemical potential, which results in the transformation of Cr^3+^ [25,51]. According to the EDS line scan analysis of the Alloy 625 segment, Ni’s content decreased but increased in O, implicating that selective dissolution occurred in Alloy 625. However, the Fe and Cr content were almost unchanged in supercritical oxidizing water, indicating stable iron oxides and chromium oxides are produced on the surface of Alloy 625 [34,36,52]. Continuously uniform corrosion layers could be observed in Alloy 625 after exposure to experiment Run-6 for 200 h, as shown in Figure 8b. In addition, the high point of O content corresponded to the high end of P content, suggesting the existence of stable phosphate in supercritical water. The same regularity mentioned above was found in 316 SS. As for TA8, a decrease in Ti content but an increase in O content. It indicates that the stable oxide of TiO_2_ or Ti_2_O_3_ could avoid severe corrosion behavior in supercritical water containing oxygen and chloride. The strength of phosphate content is also significantly low, implying that titanium-phosphorus oxides are hard to form in the high concentration of salts and oxygen.

## 4. Discussion

According to the experimental results and analysis, the general molten corrosion mechanism of the alloys about 316 SS, Alloy 625, and TA8 in a supercritical water oxidation system with a variety of inorganic salts (i.e., Cl^−^ and PO_4_^3−^) is further discussed, as shown in Figure 9.

In previous research, the oxide film growth mechanism, metal dissolution mechanism, and metal salt precipitation mechanism are typical mechanisms investigated and widely accepted by researchers [12]. Generally, the corrosion resistance of alloy materials usually depends on the stability of metal oxides on the surface. Corrosion in a variety of inorganic salts system up to supercritical temperature is determined by several solution-dependent and material-dependent factors [53].

The water dissolves in a molten salt mixture to provide an oxidant during the corrosion process. In the first stage of corrosion, the dissolved oxygen is exhausted, and the water molecules, as impurities, begin to deliver oxygen. Oxygen was absorbed into the matrix to form oxides with metal ions. An appropriate oxygen concentration will accelerate the formation of a protective oxide film, but excessive oxygen concentration might cause severe damage to the oxide layer [5]. The presence of oxygen has a significant effect on the overall solution-passivation mechanism [54].

In order to avoid further corrosion, the protective oxide films were formed at the initial corrosion stage, so the corrosion rate at the early was lower. The diffusion rate of metal ions in the alloy was also slow, and chromium formed more stable Cr_2_O_3_ to cover the surface of the material’s matrix, which can effectively prevent further corrosion. Cr_2_O_3_ is the primary oxide that protects the matrix from corrosion medium, but the Cr_2_O_3_ protective film is unstable at high temperature oxidized aqueous solution [55]. Although Cr_2_O_3_ is considered the protective oxide in acidic solutions, due to the high diffusing rate of Cr^3+^, Cr_2_O_3_ is easily converted into Cr^6+^ to further dissolve in SCWO [41]. Kritzer et al. [53] pointed out that trivalent chromium oxides are oxidized into soluble hexavalent chromium compounds at a temperature of up to 250 °C. Li et al. [56] found that chromium oxide only existed at the initial oxidation stage and disappeared after prolonged exposure. With the increase of reaction temperature and pressure, the protective oxide film formed on the alloy surface gradually dissolved. Chlorine can penetrate the matrix and destroy oxides on the surface of alloys because chlorides damage the oxide film’s completeness and react with the matrix metal. Viljoen [57] found that in molten NaCl-KCl at 727 °C, HCl would form even with the existence of little H_2_O, and iron and chromium would be chlorinated. At the same time, the products of the chlorination reaction will also react with the salt to form other compounds, which can significantly increase the corrosion rate of the materials [58]. Pessall et al. [59] found that pitting corrosion occurs in stainless steel and nickel-based alloys in oxygen-free, high-temperature phosphate solutions. The numerous corrosion holes formed on the surface of alloys with the corrosion time increases. In supercritical water containing phosphate, the diffusion of metal ions reacts with phosphate, and the product precipitates on the specimens’ surface to form a loose phosphating film.

Nevertheless, the loose phosphate oxide film is unstable and will fall off with other unstable oxides. The molten phosphate has a high ion transfer capacity. Molten phosphate offers a mobile medium and accelerates the diffusion of metal elements and aggressive ions. The reaction of the metal in the oxide layer with the soluble complex forming phosphate causes accelerated corrosion [60]. The reaction causes some of the metal cations to enter the molten phosphate, and the reaction between the metal cations and the phosphate anions produces oxides, and the resulting oxides are dissolved in the low-melting-point phosphate ion melt, and the phosphate deposit exists as eutectic salts [23]. Yishu Zhang et al. [22] observed the formation of hydrothermal molten phosphate in supercritical water and found that the melting of multiple salts during SCWO treatment of wastewater and the solvent system undergoes NaCl crystallization in the presence of phosphate.

In previous research, it was found that high-temperature electrochemical corrosion formed by corrosive anions can cause more severe corrosion in subcritical water systems. Son et al. [41] found that oxygen penetrates deeper within the alloy under supercritical conditions, while weight loss is most severe under subcritical conditions. Huang et al. [61,62] found that Alloy 625 does not easily form protective Cr films under subcritical conditions and therefore exhibits more severe corrosion. Alloy materials can suffer more severe corrosion under high-temperature subcritical conditions than under supercritical conditions [41,53]. At higher subcritical temperatures near the critical temperature, density and acid dissociation play a major contribution to corrosion, and in supercritical aqueous systems, electrochemical processes do occur as long as the density is sufficiently high [63]. Wang et al. [64] investigated the effect of H_2_O on molten salt corrosion of GH3535 alloy, and the results showed that the impurity H_2_O changed the redox state of the molten salt and enhanced its corrosion, thus accelerating the corrosion of the alloy. Although oxygen would be isolated from alloy by molten phosphate, the high-temperature electrochemical corrosion would cause severer corrosion.

## 5. Conclusions

In this research, corrosion tests of 316 SS, Alloy 625, and TA8 were performed in supercritical water (400 °C/25 MPa) containing phosphate, chloride, and oxygen. The surface and cross-section morphologies, elements distribution, phase compositions, and corrosion weight gain of the oxide films formed on 316 SS, Alloy 625, and TA8 exposed to SCW have been investigated. The morphological characteristics and elemental concentrations of the oxide films differ for the three alloys in hydrothermal molten phosphate. In hydrothermal molten phosphates, 316 SS shows the worst corrosion resistance, and TA8 demonstrates the best corrosion resistance. Stable water-insoluble phosphates are more likely to form on the surface of Alloy 625, while phosphates on the surface of 316 SS displayed the severest spallation. The corrosion products included on the surface of the 316 SS are mainly Fe_2_O_3_ and Fe_3_O_4_, and Alloy 625 are Cr_2_O_3_, NiCr_2_O_4_, and phosphates. A two-layer structure oxide film was observed on the surface of TA8 consisting of an outer layer with TiO_2_ and an inner layer with Ti_2_O_3_. The loose phosphates corrosion products formed on the surface of alloys would accelerate corrosion. Molten phosphate offers a mobile medium for accelerating the diffusion of aggressive species, and the high-temperature electrochemical corrosion would cause severer corrosion. The presence of chlorine could weaken the stability of phosphates corrosion products in supercritical water. 316 SS showed the highest corrosion rate of the alloys tested, 0.170 mmpy at the bottom of the reaction tube in the presence of molten phosphate, chlorine, and oxygen, compared to 0.203 mmpy at the top of the reaction tube, with TA8 showing the lowest corrosion rate in the corrosion experiments. At the top of the reaction tube, all three alloys show an overall mass gain. At the bottom of the tube, 316 SS and nickel-based alloy 625 show a mass loss, while TA8 alloy continues to show a mass gain. Alloy 625 showed a corrosion rate between 316 SS and TA8. Further research is needed to understand the competitive reaction between alloy and aggressive species (such as oxygen, H_2_O, chlorine, and phosphate). In addition, the mechanism of high-temperature electrochemical corrosion in supercritical water with molten phosphate is worthy of further exploration and research.

## Figures and Tables

**Figure 1 materials-16-00395-f001:**
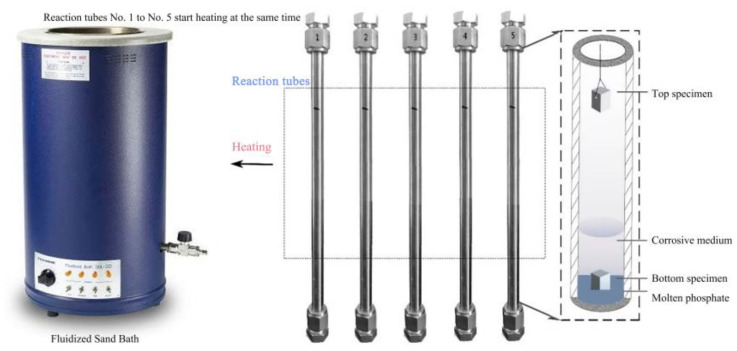
Schematic diagram of supercritical water experimental apparatus.

**Figure 2 materials-16-00395-f002:**
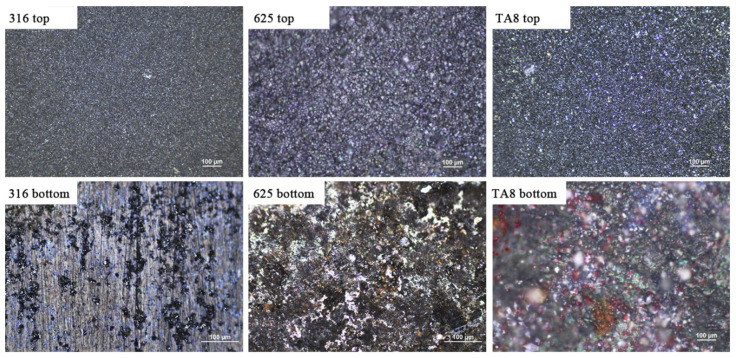
The images of 316 SS, Alloy 625, and TA8 after exposure to SCW at experiment Run-6 for 200 h.

**Figure 3 materials-16-00395-f003:**
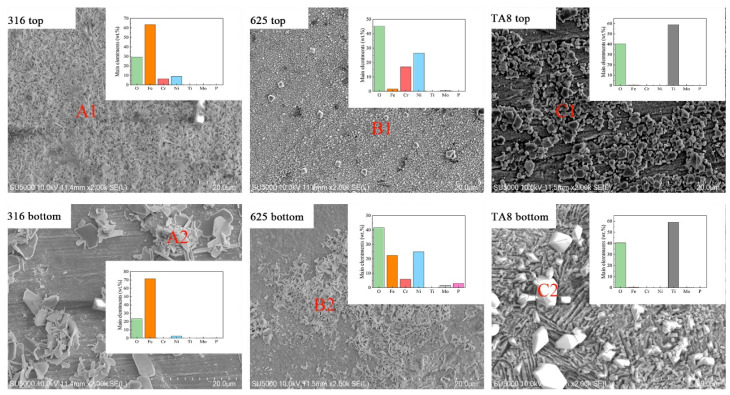
SEM images and main elements compositions of the oxide films formed on 316 SS, Alloy 625, and TA8 after exposure to SCW at experiment Run-6 for 200 h.

**Figure 4 materials-16-00395-f004:**
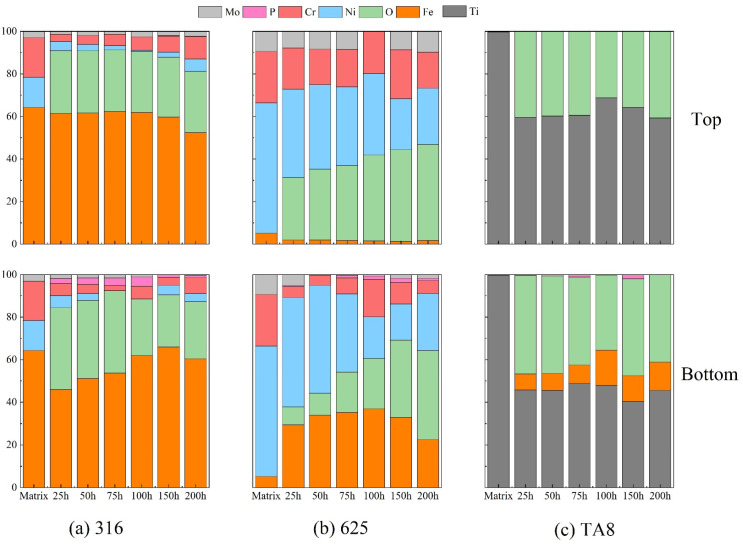
The main elements compositions of the oxide films formed on (**a**) 316 SS at the top and bottom reaction tube, (**b**) Alloy 625, (**c**) TA8 after exposure to SCW at experiments Run-1 to Run-6.

**Figure 5 materials-16-00395-f005:**
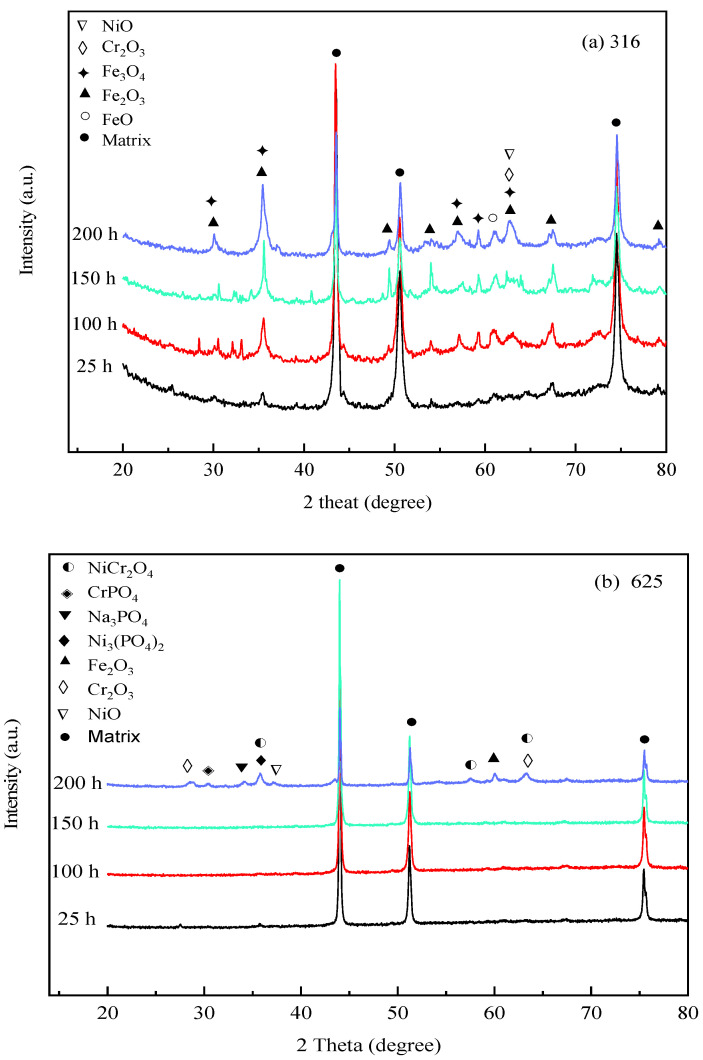
XRD spectra of the specimens after exposure to SCW at 25 h, 100 h, 150 h, and 200 h, respectively; (**a**) 316 SS at the bottom reaction tube; (**b**) Alloy 625 at the bottom reaction tube; (**c**) TA8 at the bottom reaction tube.

**Figure 6 materials-16-00395-f006:**
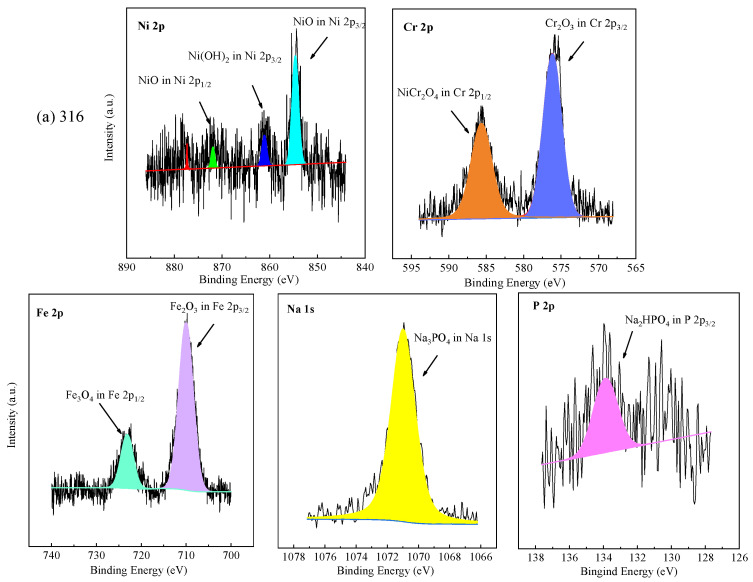
XPS patterns of the specimens at the bottom of the reaction tube after exposure to SCW for 200 h, respectively; (**a**) 316 SS at the bottom reaction tube; (**b**) Alloy 625 at the bottom reaction tube; (**c**) TA8 at the bottom reaction tube.

**Figure 7 materials-16-00395-f007:**
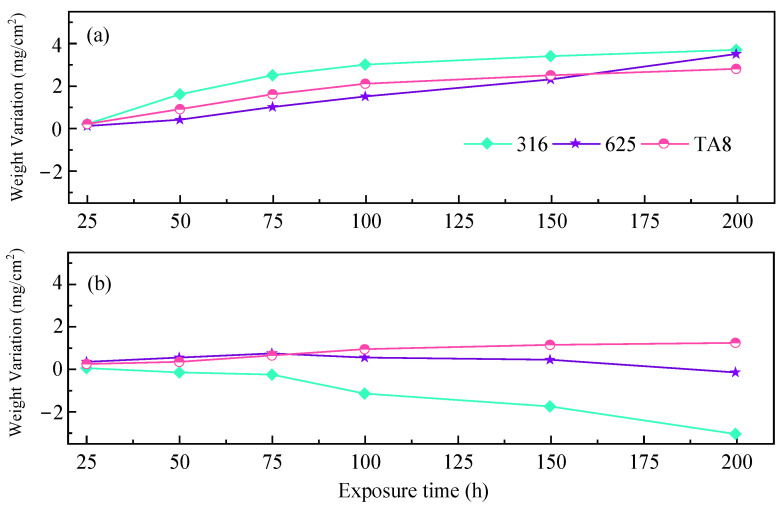
Corrosion weight gain of the tested alloys exposed to SCW at experiments Run-1 to Run-6, respectively; (**a**) the specimens at the top reaction tube; (**b**) the specimens at the bottom reaction tube.

**Figure 8 materials-16-00395-f008:**
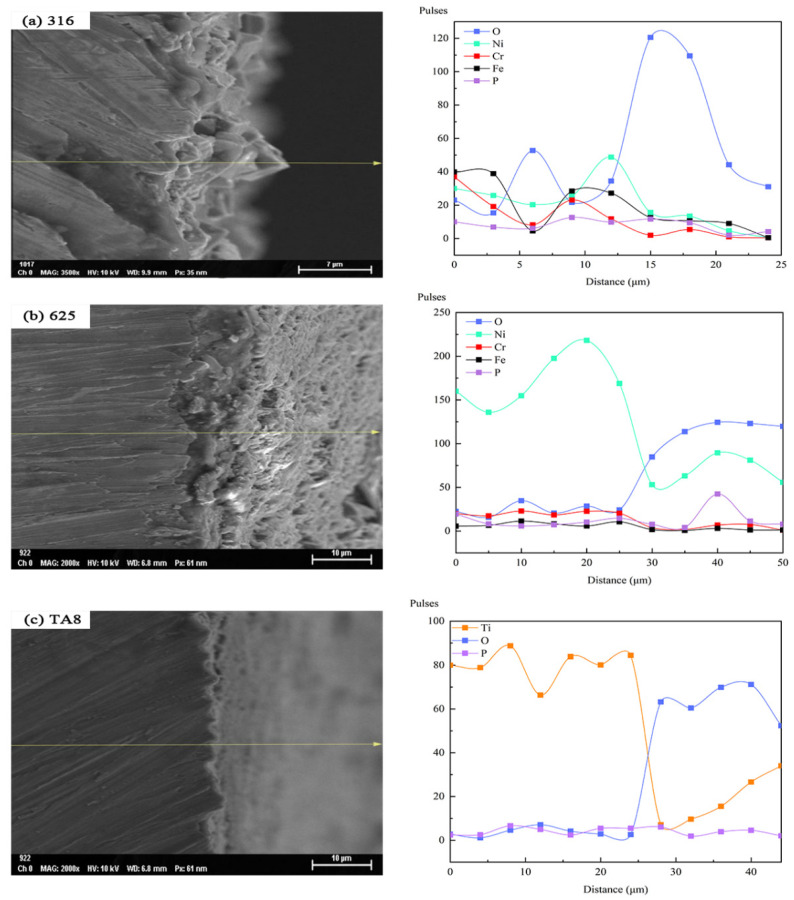
SEM morphologies and EDS line scan of the cross-section of the oxide films formed on the specimens after exposure in SCW at experiment Run-6 for 200 h, respectively; (**a**) 316 SS at the bottom reaction tube; (**b**) Alloy 625 at the bottom reaction tube; (**c**) TA8 at the bottom reaction tube.

**Figure 9 materials-16-00395-f009:**
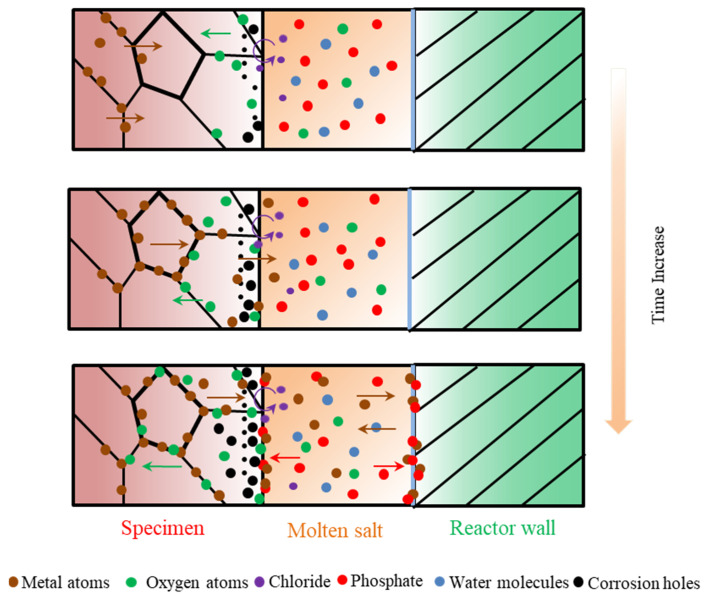
Corrosion dynamic mechanism diagram of 316 SS, Alloy 625, and TA8 exposed to supercritical water containing phosphate, chloride, and oxygen.

**Table 1 materials-16-00395-t001:** Main chemical compositions of the tested alloy (in wt.%).

Alloy	Fe	Cr	Ni	Ti	Mo	C
316 SS	64.0	18.5	14.0		3.0	0.08
Alloy 625	5.0	23.0	58.3	0.4	9.0	0.1
TA8	0.3			Bal.		0.08

**Table 2 materials-16-00395-t002:** Experimental conditions used for corrosion tests.

Experiment Run	PO_4_^3−^ (mg/L)	Cl^−^ (mg/L)	O_2_ (mg/L)	Time (h)	Temperature (°C)	Pressure (MPa)
1	4500	1500	6000	25	400	25
2	4500	1500	6000	50	400	25
3	4500	1500	6000	75	400	25
4	4500	1500	6000	100	400	25
5	4500	1500	6000	150	400	25
6	4500	1500	6000	200	400	25

## Data Availability

Not applicable.

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
