# Peer review of "The Effect of Molten Phosphate on Corrosion of 316 Stainless Steel, Alloy 625, and Titanium TA8 in Supercritical Water Oxidation Conditions"

_materials, 2023, doi:10.3390/ma16010395_

Round 1

Reviewer 1 Report

The work is ordered well with a need to use electrochemical measurements not only characterizations and weight loss. I hope to use electrochemical meaurements in your future work.

the Type of 316 Stainless steel is 316L Type or not? If yes put it as 316L stainless steel in all the manuscript.

Revise English again.

Clarify the aim in the introduction. why comparing these alloys together and they are not the same kind?

revise reference 50, where is the authors.

Author Response

Reply of authors

The authors greatly appreciate the reviewer for the suggestive and helpful comments, these comments are all valuable and helpful for revising and improving our paper, as well as the important guiding significance to our researches. The authors have carefully considered these comments and the modification have been reflected in the revised manuscript. The responds to the reviewer’s comments are as follows.

  1. Comment of reviewer

The work is ordered well with a need to use electrochemical measurements not only characterizations and weight loss. I hope to use electrochemical measurements in your future work.

Reply of authors

The authors sincerely appreciate the reviewer for this valuable suggestion. In subsequent research, we have added electrochemical measurements to the characterization.

  1. Comment of reviewer

The Type of 316 Stainless steel is 316L Type or not? If yes put it as 316L stainless steel in all the manuscript.

Reply of authors

The authors greatly appreciate the reviewer for the suggestive and helpful comment, the Type of 316 stainless steel is just 316 Stainless steel not 316L Type.

  1. Comment of reviewer

Revise English again.

Reply of authors

The authors sincerely apologize for the grammar problems existing in paper. The authors have modified the paper carefully and all modifications have been reflected in the revised manuscript and marked in yellow.

  1. Comment of reviewer

Clarify the aim in the introduction. why comparing these alloys together and they are not the same kind?

Reply of authors

Following the reviewer’s suggestion, the aim of comparing these alloys together and they are not the same kind, we have added in the introduction. “It should be noted that 316 stainless steel, nickel-based Alloy 625, and titanium alloy TA8 are often used as candidate alloys for different component areas of supercritical water oxidation systems. Three other alloys have been chosen to assess the corrosion behavior of components in various areas of supercritical water oxidation systems in a hydrothermal phosphate molten salt solvent system.”

  1. Comment of reviewer

revise reference 50, where is the authors.

Reply of authors

According to the valuable comment of reviewer, an error in the serial number of citation 50 which has been corrected and changed to 55.

Reviewer 2 Report

The manuscript is of practical and theoretical interest, however, the authors should take into account a number of remarks.

i) The abstract should be expanded to contain specific research results.

ii) Line165-166. Authors write: "Obviously, significant changes of the corrosion morphologies were observed at the conditions with or without salt " Authors should clearly articulate this difference in the manuscript.

iii) Lines 173-174: Authors write: "The compact and stable oxides film was hard to from on the surface of 316 stainless steel, resulting in the matrix was exposed to corrosive medium and the further corrosion would occur under the unprotected condition" If the film was difficult to separate from the surface, then it prevented the penetration of the medium to the matrix, and not vice versa, as the authors write.

iv) In section 4 (Discussion) it is not clear which alloy the corrosion mechanism under discussion refers to.

v) Line 450: "... the high temperature electrochemical corrosion would cause more severer corrosion".  This phrase needs clarification.

vi) Lines 457-459:"316 SS showed the highest corrosion rate of the alloys tested, 0.170 mmpy at the bottom of the reaction tube in the presence of molten phosphate, chlorine and oxygen, compared to 0.203 mmpy at the top of the reaction tube " This difference needs to be explained.

vii) The manuscript is sloppy. For example: 1) Line 152: "...at experiment Run-1 for 200 h". But Run 1 corresponds to 25 h as it is shown in Table 2. 2) Figure 2. Line 161-162: "...after exposure to SCW at experiment Run-1 for 200 h". But Run 1 corresponds to 25 h as it is shown in Table 2. 3) Line 164: ".... experiment Run-1 for 200 h," May be  experiment Run-6 for 200 h, 4) Line 310: "...exposed to the experiment Run-1 for 200 h was further... " See note above. 5) Line 370: "at experiment Run-1 for 200 h was performed, " See note above. 6) Figure 6 with XPS data is absent in the manuscript. 7) There are two Figures with number 5  and  two Figures with number 9 in the manuscript.

Author Response

Reply of authors

The authors greatly appreciate the reviewer for the suggestive and helpful comments, these comments are all valuable and helpful for revising and improving our paper, as well as the important guiding significance to our researches. The authors have carefully considered these comments and the modification have been reflected in the revised manuscript. The responds to the reviewer’s comments are as follows.

  1. Comment of reviewer

The abstract should be expanded to contain specific research results.

Reply of authors

The authors greatly appreciate the reviewer for the suggestive and helpful comment. We have expanded on the specific findings regarding corrosion rates in the abstract.

  1. Comment of reviewer

Line165-166. Authors write: "Obviously, significant changes of the corrosion morphologies were observed at the conditions with or without salt " Authors should clearly articulate this difference in the manuscript.

Reply of authors

According to the valuable comment of reviewer, we have added a description of the differences in the main results.

  1. Comment of reviewer

Lines 173-174: Authors write: "The compact and stable oxides film was hard to from on the surface of 316 stainless steel, resulting in the matrix was exposed to corrosive medium and the further corrosion would occur under the unprotected condition" If the film was difficult to separate from the surface, then it prevented the penetration of the medium to the matrix, and not vice versa, as the authors write.

Reply of authors

The authors sincerely apologize for the grammar problems existing in paper, we have corrected the grammar of the above sentence. The sentence "The compact and stable oxides film was hard to from on the surface of 316 stainless steel, resulting in the matrix was exposed to corrosive medium and the further corrosion would occur under the unprotected condition." have been modified into "The compact and stable oxide film was hard to form on the surface of 316 stainless steel, resulting in the matrix being exposed to a corrosive medium, and further corrosion would occur under unprotected conditions.".

  1. Comment of reviewer

In section 4 (Discussion) it is not clear which alloy the corrosion mechanism under discussion refers to.

Reply of authors

The authors sincerely appreciate the reviewer for this valuable suggestion. We have modified the corrosion mechanism for the three alloys in molten phosphate. The sentence is "the general molten corrosion mechanism of the alloys about 316 stainless steel, Inconel 625, and TA8 in a supercritical water oxidation system with a variety of inorganic salts. "

  1. Comment of reviewer

Line 450: "... the high temperature electrochemical corrosion would cause more severer corrosion".  This phrase needs clarification.

Reply of authors

The authors sincerely appreciate the reviewer for this valuable suggestion. We add a discussion of the causes of more severe corrosion due to high temperature electrochemical corrosion. "Nevertheless, the loose phosphate oxide film is unstable and will fall off with other unstable oxides. The molten phosphate has a high ion transfer capacity, molten phosphate offers a mobile medium and accelerates the diffusion of metal elements and aggressive ions. The reaction of the metal in the oxide layer with the soluble complex forming phosphate causes accelerated corrosion. The reaction causes some of the metal cations to enter the molten phosphate, and the reaction between the metal cations and the phosphate anions produces oxides, and the resulting oxides are dissolved in the low-melting point phosphate ion melt, and the phosphate deposit exists as eutectic salts. Yishu Zhang et al. observed the formation of hydrothermal molten phosphate in supercritical water, found that the melting of multiple salts during SCWO treatment of wastewater and the solvent system undergoes NaCl crystallization in the presence of phosphate. In previous researchs, it was found that high-temperature electrochemical corrosion formed by corrosive anions can cause more severe corrosion in subcritical water systems. Son et al. found that oxygen penetrates deeper within the alloy under super-critical conditions, while weight loss is most severe under subcritical conditions. Huang et al. found that Inconel 625 does not easily form protective Cr films under subcritical conditions, and therefore exhibits more severe corrosion. Alloy materials can suffer more severe corrosion under high temperature subcritical conditions than under supercritical conditions. At higher subcritical temperatures, near the critical temperature, density and acid dissociation play a major contribution to corrosion, and in supercritical aqueous systems, electrochemical processes do occur as long as the density is sufficiently high. Wang et al. investigated the effect of H2O on molten salt corrosion of GH3535 alloy, and the results showed that the impurity H2O changed the redox state of the molten salt and enhanced its corrosion, thus accelerating the corrosion of the alloy. Although oxygen would be isolated from alloy by molten phosphate, the high-temperature electrochemical corrosion would cause more severer corrosion. "

  1. Comment of reviewer

Lines 457-459:"316 SS showed the highest corrosion rate of the alloys tested, 0.170 mmpy at the bottom of the reaction tube in the presence of molten phosphate, chlorine and oxygen, compared to 0.203 mmpy at the top of the reaction tube " This difference needs to be explained.

Reply of authors

According to the reviewer’s helpful suggestion, we have added the explain in the Manuscript.  " The corrosion rate of the specimen located at the bottom is lower than that of the top, because in terms of mass change, the overall mass of the bottom specimen shows weight loss, and the top specimen shows an overall increase in mass, which is caused by the flaking of the oxide film on the surface of the specimen due to hydrothermal molten phosphate corrosion. "

  1. Comment of reviewer

The manuscript is sloppy. For example: 1) Line 152: "...at experiment Run-1 for 200 h". But Run 1 corresponds to 25 h as it is shown in Table 2. 2) Figure 2. Line 161-162: "...after exposure to SCW at experiment Run-1 for 200 h". But Run 1 corresponds to 25 h as it is shown in Table 2. 3) Line 164: ".... experiment Run-1 for 200 h," May be  experiment Run-6 for 200 h, 4) Line 310: "...exposed to the experiment Run-1 for 200 h was further... " See note above. 5) Line 370: "at experiment Run-1 for 200 h was performed, " See note above. 6) Figure 6 with XPS data is absent in the manuscript. 7) There are two Figures with number 5  and  two Figures with number 9 in the manuscript.

Reply of authors

The authors sincerely apologize for the grammar problems existing in paper. This was caused by a mistake in our work and we have supplemented the article with XPS graphics and corrected the ordering of the images and run numbers.

Reviewer 3 Report

The topic covered in the paper is interesting and the methodology adopted is robust. The results are clear and of scientific relevance. However, some clarifications and modifications are needed:

General comments

- Improve your English.

- Check the use of acronyms.

- Check the number of the figures in the captions and in the manuscript.

Introduction

- A reference is missing in the sentence "Ma et al. [...]".

Methods

- What is the size of the hole? Can this feature affect the results?

- Fig. 1: you test plates but in the figure there are cubes. How is the bottom plate arranged?

Results

- In the whole section it is not specified which surface of the specimens is analyzed. Is the corrosion process homogeneous on the specimens?

Author Response

Reply of authors

The authors greatly appreciate the reviewer for the suggestive and helpful comments. These comments are all valuable and helpful for revising and improving our paper and the essential guiding significance of our research. The authors have carefully considered these comments, and the modification has been reflected in the revised manuscript. The responses to the reviewer's comments are as follows.

  1. Comment from the reviewer

General comments

- Improve your English.

- Check the use of acronyms.

- Check the number of the figures in the captions and in the manuscript.

Reply of authors

The authors sincerely apologize for the grammar problems existing in the paper. We checked the use of acronyms and the number of figures in the captions and the manuscript again. The authors have modified the paper carefully, and all modifications have been reflected in the revised manuscript and marked in yellow.

  1. Comment from the reviewer

Introduction

- A reference is missing in the sentence "Ma et al. [...]".

Reply of authors

According to the valuable comment of the reviewer, we have added the missing references.

  1. Comment from the reviewer

Methods

- What is the size of the hole? Can this feature affect the results?

- Fig. 1: you test plates but in the figure there are cubes. How is the bottom plate arranged?

Reply of authors

The authors greatly appreciate the reviewer for the suggestive and helpful comment. The diameter of the hole is 1 mm because the hole area is tiny compared to the specimen and does not affect the experimental results. We used a wire cut to cut a small hole from the edge of the specimen. In the experiments, we used plate specimens to obtain a better corrosion mass change curve. In drawing the schematic diagram in Figure 1, the dimensions are to scale, but due to the visual effect of different angles are finally presented as a cube. Ceramic spacers are used on the bottom specimen to prevent direct contact with the bottom of the reaction tube.

  1. Comment from the reviewer

Results

- In the whole section it is not specified which surface of the specimens is analyzed. Is the corrosion process homogeneous on the specimens?

Reply of authors

The authors sincerely appreciate the reviewer for this valuable suggestion. In the profile analysis, the bottom surface of the specimen was used, which has been revised in the text. Because of a recessed circular gasket, the bottom of the specimen can be in complete contact with the corrosion medium, and the corrosion process is uniform.

Reviewer 4 Report

The study is interesting. However, a minor revision is necessary before acceptance.

Comments and Suggestions for Authors

1. The introduction session is too long for an article.

2. In the results session, the study regarding the corrosion rate does not appear, although the results for 316 stainless steel appear in the conclusions. Specify how you determined these values.

2. The discussion section must discuss the results obtained with the current literature.

3. The conclusions should be more objective, they are not coherent.

At the end of the conclusions, present the corrosion rate for 316 SS, but not for the other 2 alloys. It is important for the comparison to present all the corrosion rates.

Author Response

Reply of authors

The authors greatly appreciate the reviewer for the suggestive and helpful comments, these comments are all valuable and helpful for revising and improving our paper, as well as the important guiding significance to our researches. The authors have carefully considered these comments and the modification have been reflected in the revised manuscript. The responds to the reviewer’s comments are as follows.

  1. Comment of reviewer

The introduction session is too long for an article.

Reply of authors

Following the reviewer’s valuable comments and suggestions, we have streamlined the introduction section again and reduced the number of words in the introduction section.

  1. Comment of reviewer

In the results session, the study regarding the corrosion rate does not appear, although the results for 316 stainless steel appear in the conclusions. Specify how you determined these values.

Reply of authors

According to the valuable comment of reviewer, we have indicated in the results how to determine these values, which we calculated using the mass change based on the uniform corrosion rate equation.

  1. Comment of reviewer

The discussion section must discuss the results obtained with the current literature.

Reply of authors

The authors greatly appreciate the reviewer for the suggestive and helpful comment. We have revised the discussion of the results obtained in the current literature and have made content and literature additions.

  1. Comment of reviewer

The conclusions should be more objective, they are not coherent. At the end of the conclusions, present the corrosion rate for 316 SS, but not for the other 2 alloys. It is important for the comparison to present all the corrosion rates.

Reply of authors

The authors sincerely appreciate the reviewer for this valuable suggestion. We have modified the concluding statements to make them more coherent, added corrosion rates for the other two alloys and added comparative notes.
